# Exosomal DNA: Role in Reflecting Tumor Genetic Heterogeneity, Diagnosis, and Disease Monitoring

**DOI:** 10.3390/cancers16010057

**Published:** 2023-12-21

**Authors:** Ziyi Xiang, Qihui Xie, Zili Yu

**Affiliations:** 1State Key Laboratory of Oral & Maxillofacial Reconstruction and Regeneration, Key Laboratory of Oral Biomedicine Ministry of Education, Hubei Key Laboratory of Stomatology, School & Hospital of Stomatology, Wuhan University, Wuhan 430079, China; 2020303042013@whu.edu.cn; 2Department of Oral and Maxillofacial Surgery, School and Hospital of Stomatology, Wuhan University, Wuhan 430079, China

**Keywords:** liquid biopsy, extracellular vesicles, exosomal DNA, biomarker

## Abstract

**Simple Summary:**

Liquid biopsy is an emerging non-invasive biopsy technique. Compared to traditional tissue biopsy, it overcomes the challenges of tumor heterogeneity, allows for multiple sampling, dynamic real-time monitoring, and offers high sensitivity and specificity. Exploring stable and sensitive effective biopsy biomaterials is crucial. The objective of our research is to evaluate the biomarker role of DNA in exosomes, as exosomes demonstrate higher stability and detectability compared to currently used biopsy materials, but there is still a lack of research on exosomal DNA. We explored the formation and cargo loading mechanism of exosomes, summarized the research results on the relationship between mitochondrial DNA and nuclear DNA in exosomes with clinical tumors so far, and proposed several suggestions for the clinical application of exosomal DNA. Exosomal DNA has demonstrated valuable biomarker roles for certain tumors, providing guidance for clinical diagnosis, treatment, and disease monitoring.

**Abstract:**

Extracellular vesicles (EVs), with exosomes at the forefront, are key in transferring cellular information and assorted biological materials, including nucleic acids. While exosomal RNA has been thoroughly examined, exploration into exosomal DNA (exoDNA)—which is stable and promising for cancer diagnostics—lags behind. This hybrid genetic material, combining contributions from both nuclear and mitochondrial DNA (mtDNA), is rooted in the cytoplasm. The enigmatic process concerning its cytoplasmic encapsulation continues to captivate researchers. Covering the entire genetic landscape, exoDNA encases significant oncogenic alterations in genes like TP53, ALK, and IDH1, which is vital for clinical assessment. This review delves into exosomal origins, the ins and outs of DNA encapsulation, and exoDNA’s link to tumor biology, underscoring its superiority to circulating tumor DNA in the biomarker arena for both detection and therapy. Amidst scientific progress, there are complexities in the comprehension and practical application of the exoDNA surface. Reflecting on these nuances, we chart the prospective research terrain and potential pitfalls, forging a path for future inquiry. By illuminating both the known and unknown facets of exoDNA, the objective of this review is to provide guidance to the field of liquid biopsy (LB) while minimizing the occurrence of avoidable blind spots and detours.

## 1. Introduction

Recently, a new biopsy method called liquid biopsy (LB) has gained attention for its ability to detect various fluid components and screen for early stage tumors and evaluate drug efficacy. The main materials used in LB for tumor diagnosis are tumor circulating cells (CTCs), circulating tumor DNA (ctDNA), and extracellular vesicles (EVs), which contain tumor-specific biomarkers such as DNA driver mutations, miRNA, and cell molecules [1,2,3,4,5]. However, there are some issues with ctDNA. It mainly comes from serum-free ctDNA and is susceptible to degradation by nucleases, making it difficult to detect. Additionally, interference from other free DNA can also pose a challenge. As a result, disease diagnosis and treatment may be delayed until ctDNA secretion reaches a detectable level, which typically occurs in the advanced stages of a tumor. Therefore, the effective acquisition and detection of tumor-derived DNA is an area of focus [3,4,5].

In recent years, the value of EVs in LB has become increasingly evident [6]. EVs are small lipid membrane vesicles that are produced by almost all cells [7], and tumor cells secrete more of them [8]. They play a critical role in intercellular communication, as well as tumor growth, migration, and recurrence [1,9]. EVs carry various bioactive molecules, including DNA, miRNA, lipids, and proteins, and their genetic characteristics are consistent with the parental cells, reflecting the physiological and pathological status of the body [2,3,4]. Of the various molecules, exosomal DNA (exoDNA) is particularly noteworthy, comprising nuclear DNA, mitochondrial DNA (mtDNA), and even viral DNA [10]. This genetic material can be found either on the surface or within vesicles [5,11,12]. Nuclear DNA and mtDNA serve as representatives of the entire human genome and act as biomarkers in various cancer-related contexts. Due to the protective effect of the vesicle membrane, the nucleic acid cargo it carries is less susceptible to degradation by nucleases, making it more stable and detectable than ctDNA [5]. Therefore, it is usually more effective to analyze exoDNA than ctDNA, and there is a higher detection sensitivity [10]. This makes exoDNA a more advantageous biopsy material compared to free ctDNA, as shown in Figure 1.

In recent decades, the diagnostic and prognostic value of miRNA and protein molecules in EVs as tumor biomarkers has been extensively studied [4,5,13,14,15]. For instance, serum exosomal miR-1247-3p levels are associated with lung metastasis in patients with hepatocellular carcinoma (HCC), miR-210 is associated with tumorigenesis in clear cell renal cell carcinoma; the increase in circulating exosomal miR-21 levels is correlated with a range of cancers including glioblastoma, pancreatic, colorectal, liver, breast, ovarian, and esophageal cancers [15]. Furthermore, there is a notable association between heightened urinary exosomal miR-21 concentrations and both bladder and prostate cancers. Other microRNAs, such as miR-155, members of the miR-17-92 cluster, and miR-1246, are reportedly overexpressed in various malignancies like brain, pancreatic, colorectal, liver, breast, and prostate cancers, as well as in oral squamous cell carcinoma, lymphoma, and leukemia. Additionally, alterations in phosphorylated and membrane-bound proteins in circulating exosomes show promise as diagnostic indicators for breast cancer. Glypican-1(GPC1) is specifically recognized for aiding in the detection of pancreatic, breast, and colorectal cancers. Meanwhile, the surface protein CD147 is implicated in the prognosis of colorectal cancer, and the particular makeup of phosphatidylserine (PtdSer) may hold significance for the early diagnosis of breast and pancreatic tumors, among others.

However, the medical significance of exoDNA as a tumor biomarker is still in its early stage, with ongoing research focused on investigating pathogenic mutations in exoDNA and its potential in tumor diagnosis [12,16,17,18,19,20]. This review aims to consolidate the existing knowledge on the role of exoDNA in tumor diagnosis and therapy. We specifically emphasize the clinical relevance of exoDNA in various tumor types, examine the range of detection methods available for exoDNA, and analyze the advantages and challenges associated with its integration into LB approaches. Our objective is to provide insights that could enhance the accuracy of future LB techniques. Our review also aims to highlight the promise of exoDNA as a pivotal tumor biomarker and stimulate further research in this field. By comprehending the clinical implications of exoDNA and overcoming technical obstacles, we envision significant advancements in refining LB methodologies for the diagnosis and treatment of tumors.

## 2. Formation of EVs and Mechanisms of exoDNA Loading

Based on their size and biogenetic pathways, EVs can be delineated into three distinct types: exosomes, microvesicles, and apoptotic bodies (ABs) [4,17,21]. Each subtype of EV possesses unique characteristics and functions in intercellular communication. The biogenesis and cargo loading mechanisms also vary among these subtypes. Exosomes are smaller in size (30–150 nm). Their biogenesis is multifaceted and originates from endocytosis. As depicted in Figure 2, their formation initiates with the inward budding of the plasma membrane (PM), giving rise to early endosomes [22]. This endosomal network becomes more complex as early endosomes mature into late endosomes, which in turn undergo inward budding to generate multivesicular bodies (MVBs) filled with intraluminal vesicles (ILVs). The culmination of this process occurs when MVBs merge with the PM, releasing ILVs as exosomes into the extracellular environment [5,23,24]. The orchestration of this intricate process leverages multiple proteins, including the endosomal sorting complexes required for transport (ESCRT), vacuolar ATPase, and Vps4, which are instrumental in segregating ubiquitinated proteins and directing them into ILVs [25,26]. Microvesicles are larger vesicles with dimensions typically spanning 100–1000 nm. These emerge when the PM of either living or effete cells buds outward [27] and eventually detaches. Conjuring processes similar to those involved in exosome production, the generation of microvesicles similarly requires the involvement of various protein factors, including Ca^2+^-dependent aminophospholipid translocases (flipases and flopases), sphingomyelinase 2 (nSMase2), scramblases, and calpain. These proteins play a pivotal role in phospholipid reorganization, membrane curvature, and reconstitution of the actin cytoskeleton, all of which contribute to the detachment and release of microvesicles into the extracellular space [18]. ABs are the bulkiest class of vesicles, with diameters ranging from 50 to 2000 nm [28]. These vesicles form during the advanced stages of apoptosis and emanate from membrane blebbing. Throughout this stage, expiring cells extrude vesicles laden with fragmented DNA, a hallmark of late-stage programmed cell death [29].

The loading mechanism of DNA into exosomes is still a subject of ongoing research, and there is no consensus in the field yet. A prominent theory proposed by Yokoi is the micronucleus (MN) theory [24], in which tetraspanins play an important role. As shown in Figure 2, MN is a cytoplasmic structure surrounded by the nuclear membrane, which forms due to the incorrect segregation of nuclear substances such as chromosomes during cell division. Both MN and nuclear exosomes have a high nuclear content and similar protein composition, and these structures have been observed to actively interact within living cells. The MN membrane is inherently unstable and eventually ruptures, leading to the release of its contents, including genomic DNA (gDNA), into the cytoplasm [23]. These nuclear materials are then enveloped by MVBs and loaded into EVs [24]. During this process, tetraspanins like CD63 can promote the formation of complexes between gDNA and nuclear proteins, which is very important for the loading of DNA into exosomes [30]. However, further research is needed to fully elucidate the complex substrates and the role of functional protein molecules involved in the exoDNA loading process.

## 3. Research Progress on the Application of exoDNA in LB

exoDNA plays a crucial role as a representative of the entire genome and reflects the mutation status of parental tumor cells. The majority of DNA within tumor-derived EVs is double-stranded DNA (dsDNA), which is protected by membranes and exhibits remarkable stability [12]. This dsDNA can be easily separated and enriched from complex plasma samples by utilizing surface markers on EVs [31]. As a result, DNA derived from tumor-derived EVs holds significant potential as a biomarker for the early detection of cancer and for monitoring the treatment response. Moreover, the exoDNA present in tumor-derived EVs has translational value, as the downstream miRNAs and proteins it encodes by this DNA may serve as therapeutic targets or predictive biomarkers for specific diseases [21]. These findings suggest that exoDNA in tumor-derived EVs not only provides valuable insights into the genetic makeup and mutational status of tumors but also offers promising avenues for the development of targeted therapies and personalized medicine [32]. This approach eliminates the need for invasive and inaccurate direct tumor sampling, presenting a new biopsy method to enhance tumor diagnosis and treatment. Overall, the utilization of exosomal nucleic acids, particularly exoDNA, as robust tools in various applications in cancer diagnosis and treatment showcases its immense potential [33]. It enables non-invasive and accurate assessment of the tumor genome, paving the way for improved cancer management strategies.

### 3.1. MtDNA as a Potential Biomarker for Tumor Diagnosis and Treatment

Mitochondria, pervasive and essential organelles within eukaryotic cells, serve as the powerhouse for energy and biomolecule synthesis and are key players in processes such as cell division and programmed cell death. Much like nuclear DNA, mtDNA can be encapsulated and transported in EVs [34]. EVs are particularly remarkable as they can harbor an entire mitochondrial genome. This enables the transfer of mtDNA to metabolically impaired cells, potentially rejuvenating their metabolic functions. Interestingly, research has uncovered that such horizontal mtDNA transfer may reawaken dormant cancer cells, potentially conferring them with resistance to chemotherapy [31]. Additionally, studies have documented that both normal and cancerous cells in culture can release intact mitochondria that are fully capable of respiration [32]. Findings indicate that alterations in the mtDNA copy number may be linked to a spectrum of human disorders, from cancer to neurodegenerative and cardiovascular diseases [35]. Hence, the presence of mtDNA within tumor-derived EVs offers an innovative circulating free DNA (cfDNA) biomarker for the surveillance and monitoring of tumors, aiding in our understanding of tumor biology and the development of targeted therapeutic strategies. The known mtDNA mutations in exosomes and their clinical utility results are shown in Table 1. 

#### 3.1.1. Prostate Cancer (PCa)

In a study conducted by Julie et al. [36,37,38], next-generation deep sequencing of the complete mitochondrial genome (16,569 bp) was performed on six cases of primary PCa tumors. The researchers discovered frequent somatic mtDNA mutations targeting respiratory complex I (RCI), totaling 12 mutations. Among them, 67% (8/12) were found to be in RCI. Protein expression of RCI was significantly decreased (*p* < 0.0002) in the primary tumors of patients with the deletion mutation at nucleotide position 11,038 (mt-nd4, 11038delA), which encodes complex I of the mitochondrial genome. Furthermore, PCa cells with mtDNA deletion mutations exhibited increased mitochondrial fusion, indicating that these genetic alterations induce mitochondrial fusion in human PCa cells [37]. Animal experiments using Hi-myc mice, which exhibited a similar pattern of mtDNA mutations to humans, revealed that Hi-myc mice carrying the mutation showed disrupted mitochondrial integrity. In this study, in 100% of cases, the researchers detected RCI mtDNA (MT-ND4) in EVs isolated from the serum of men with benign prostatic hyperplasia (5/5), men with a Gleason score of 6 PCa (5/5), and men with a Gleason score of 9 PCa (5/5). The EV population obtained from the serum of these cases contained MFN2 and IMMT proteins, which are associated with mitochondrial fusion and biogenesis. This finding suggests that RCI-specific mtDNA and mt-targeted proteins can be detected in EVs in the serum of both benign and malignant PCa patients [39]. The researchers hypothesized that EVs migrate to mitochondria and capture mtDNA and mt proteins [24]. This hypothesis was confirmed by the co-localization of mitochondria and EVs using confocal imaging, supporting the idea that EVs may have migrated to the mitochondria [24,36,37,38,39].

Additionally, James et al. [42] developed the EPI test, which identifies the presence of >Gleason grade group 2 (GG2) type PCa in men over 50 years old with an initial biopsy prostate-specific antigen (PSA) level of 2–10 ng/mL [40,41]. The test detects the expression of three genes in urinary exosomes. In comparison to patients with GG1 and benign disease, an EPI score > 15.6 indicates a high risk for GG2 type PCa [38]. This test is aimed at reducing unnecessary biopsies and treatments as a result of low PSA specificity [43].

#### 3.1.2. Breast Cancer

Currently, there is limited research on the evolution of treatment for ER^+^ breast cancer from hormone therapy sensitivity (HTS)/responsiveness (HTD) to the development of hormone therapy resistance (HTR) [47,48,49]. However, a study conducted by Pasquale et al. [46] provided additional insights in this area. The study found that hormone treatment induced defects in OXPHOS of breast cancer cells. As a result, these cells were able to restore their vitality through the transfer of mtDNA carried by EVs. This transfer of mtDNA primarily occurred in cancer stem cell-like cells, ultimately leading to resistance to hormone therapy in breast cancer [44]. To further investigate this phenomenon, the researchers established a xenograft model of HTR metastatic disease. They observed the presence of EVs carrying mtDNA in the peripheral circulation. They then performed the experimental introduction of breast cancer EVs using HTR samples in mice and quantitatively determined the copy numbers of mtDNA. The findings of the study demonstrated that EVs carrying high levels of mtDNA promoted the reawakening of intracavitary breast cancer cells from a metabolically dormant state induced by hormone therapy [45,46]. In vivo, the transfer of mtDNA predominantly occurs in cancer stem cell-like cells, and the acquisition of mutant mtDNA takes place during the transition from HTS to HTR tumors. Based on these findings, it can be concluded that the transfer of mtDNA carried by EVs acts as a carcinogenic signal, enabling hormone therapy-induced cancer stem cell-like cells to overcome dormancy and leading to resistance against endocrine therapy in OXPHOS-dependent breast cancer [46].

These findings provide valuable insights into the mechanisms underlying the development of HTR and suggest potential targets for future therapeutic interventions in ER^+^ breast cancer.

#### 3.1.3. Glioblastoma

Research conducted by Beata et al. [60] sheds light on the potential of using the mtDNA copy number as a biomarker for glioblastoma tumorigenesis. Glioblastoma, the most prevalent malignant tumor in the adult central nervous system, is characterized by increased glucose consumption and associated with a poor prognosis [50,51,52]. Due to the limited non-invasive access to the brain, the use of exoDNA for detection is of great clinical value [53,54,55]. In this study, mtDNA copy numbers and point mutations were evaluated in tissue and exoDNA samples from glioblastoma patients and a control group. The findings demonstrate that the average mtDNA copy number in brain tissue and EVs was lower in the glioblastoma patient group compared to the control group (K-W test, *p* < 0.01). While the frequency of mtDNA point mutations showed slight variations between cases and controls, the clinical implications of these findings were inconclusive. Considering that the glioblastoma was characterized as the Warburg phenotype [56,57], the researchers suggested that the observed mtDNA mutations and copy number changes may have contributed to the adaptation of altered metabolic pathways [58]. This hypothesis is supported by the discovery of mtDNA variations in Beata’s study, specifically in the mt-12S rRNA region, which can impact the oxidative capacity of the respiratory chain. However, it is important to note that further research is required to fully understand the potential of mtDNA as a reliable biomarker for glioblastoma [54,55]. These findings provide valuable insights into the molecular characteristics of glioblastoma and highlight the potential use of mtDNA as a biomarker for this aggressive brain tumor [59]. Further studies will help validate these findings and explore the clinical implications, ultimately contributing to the development of improved diagnostic and therapeutic strategies for glioblastoma patients.

#### 3.1.4. Ovarian Cancers

In a study conducted by Judit et al. [61,62], exoDNA was extracted from blood and plasma samples of 24 patients with serous epithelial ovarian cancers and 24 healthy control subjects. The researchers utilized real-time quantitative PCR to measure the copy numbers of mtDNA. The study found that the copy numbers of mtDNA in exosomes was significantly higher in tumor patients and advanced cancer patients, as well as individuals with FIGO stage III and IV, compared to FIGO stage I patients. This suggests a notable disparity in wb-mtDNA copy number between patients with early stage and advanced-stage cancer patients. Moreover, the research revealed that exosomes exhibited the highest mtDNA copy numbers, followed by the plasma and peripheral blood of patients with advanced cancer. Conversely, healthy individuals and patients with early stage cancer showed the opposite pattern. The increased levels of both free-cell and exosomal mtDNA in advanced cancer align with the characteristics of tumor-derived DNA promoting metastasis [65]. It can be concluded that an altered mtDNA copy number might serve as a signal for cancer progression in ovarian cancer [62,63,64]. Further research will be needed to validate these findings and explore the clinical utility of mtDNA copy number as a biomarker in ovarian cancer patients [66].

### 3.2. Known Relationships between Nuclear DNA Mutations and Clinical Tumors

Previous research has suggested that analyzing the genomic content of exosomes can be a promising approach for studying the mutation landscape associated with cancer, providing a potential avenue for early detection methods [5,10,12,17,67,68]. Exosomes not only contain mtDNA but also carry a large amount of nuclear DNA, similar to the gDNA found in cell nuclei, evenly distributed across all chromosomes; this further reveals their significance and potential influence in various biological processes [24,31,69]. This includes information about copy number profiles, deletions, gene fusions, point mutations, insertions, and mutational signatures [6]. By analyzing the genetic material of exoDNA, researchers can non-invasively obtain critical information on tumor mutations, enabling the detection of early stage cancers and facilitating the monitoring of disease progression and treatment outcomes, thereby presenting new opportunities for conducting LB in patients [70]. Further research is needed to optimize and standardize the techniques for exoDNA analysis, as well as to validate its clinical utility as a diagnostic tool, offering new possibilities for precision medicine and personalized treatment strategies. The known nuclear DNA mutations in exosomes and their clinical utility results are shown in Table 2.

#### 3.2.1. Pancreatic Ductal Adenocarcinoma (PDAC)

PDAC is often characterized by mutations in the KRAS and p53 genes. Researchers performed Sanger sequencing on PCR-amplified DNA obtained from exosomes derived from the Panc-1 human pancreatic cancer cell line [69]. They successfully detected KRAS and p53 mutations in the exoDNA and discovered that the majority of circulating DNA in serum samples may originate from exosomes rather than existing as free-floating DNA. Specific mutations included a base exchange from GGT to TGT on codon 12 in KRAS and a base change from CAG to CTG on codon 22 in KRAS. Additionally, in a study by Sonia et al. [71], a specific cell surface proteoglycan called GPC1 was identified to be abundantly present in tumor-derived exosomes. These GPC1-positive circulating exosomes (GPC1^+^ crExos) were found to carry specific KRAS mutations. The detection of GPC1^+^ crExos in the serum of pancreatic cancer patients showed exceptional specificity and sensitivity, allowing for clear differentiation between healthy individuals, patients with benign pancreatic diseases, and those with early and advanced-stage pancreatic cancer. Moreover, the diagnostic efficacy of GPC1^+^ crExos surpassed that of the traditional marker CA19-9 and MRI. Importantly, the levels of GPC1^+^ crExos were found to correlate with tumor burden and patient survival both before and after surgery. These findings offer a more precise and sensitive approach compared to existing methods, providing new possibilities for personalized treatment strategies and improved patient management in pancreatic cancer cases.

In a comprehensive study, a large cohort of PDAC patients was analyzed by examining the genomic content of plasma exoDNA [70,72]. The researchers utilized quantitative droplet digital PCR (ddPCR) to specifically target KRAS gene mutations in both exoDNA and ctDNA. The results revealed distinct patterns in the KRAS mutation allele frequency (MAF) across different sample types. Baseline metastatic samples exhibited the highest average MAF, followed by localized disease and cystic lesions, with non-tumor pancreatic diseases demonstrating the lowest MAF. Importantly, the combined detection of ctDNA or exoDNA MAF along with the commonly used biomarker CA19-9 provided a complementary prognostic value [6,73,74,75]. Additionally, a significant correlation was observed between exoDNA KRAS MAF and the patients’ radiological progression over time: patients with an exoDNA KRAS MAF of 1% had a 100% probability of disease progression during treatment, while those with an MAF below 1% had a 90% probability of non-progression. Moreover, continuous LB in patients with locally advanced PDAC possess therapeutic predictiveness: the reduction in exosomal KRAS MAF after completing neoadjuvant therapy, compared to baseline levels, is correlated with the likelihood of surgical resection.

These findings underscore the potential of longitudinal monitoring of exoDNA to obtain unique predictive information on the outcomes of adjuvant therapy for localized disease and the metastatic progression of PDAC, which may hold valuable clinical significance for therapeutic decision making and improving patient prognosis.

#### 3.2.2. Gliomas

Recent research has shed light on the relationship between cfDNA and tumor invasiveness in patients with gliomas [76,77]. Research has found that there are varying concentrations of exoDNA in plasma at different stages of tumor growth [78]. In low-grade gliomas characterized by a low mitotic index, small tumor volume, and absence of necrotic areas, circulating exoDNA exhibited a direct correlation with the overall tumor volume. On the other hand, high-grade gliomas showed a negative correlation between the concentration of exoDNA in plasma and tumor volume, mitotic index, and Ki-67 [79]. The correlation between exoDNA concentration and tumor grade may be influenced by various factors, including tumor angiogenesis, growth, metastasis, and evasion of immune surveillance. These findings suggest that exoDNA could potentially be used for early detection of high-grade glioma recurrence and identification of asymptomatic low-grade gliomas in the diagnostic process. By measuring exoDNA concentration, healthcare professionals may be able to detect gliomas at an early stage, provide timely intervention, and monitor tumor progression. This could potentially improve patient outcomes and contribute to more effective management of gliomas.

A study by Noemí et al. [80] showed that different types of EVs released by glioblastoma cells, including ABs, microvesicles, and exosomes, have the ability to traverse the intact blood–brain barrier. These EVs carry DNA sequences in the peripheral blood of brain tumor patients, which can effectively be utilized to detect biomarkers such as IDH1 [81,82], regardless of the blood–brain barrier’s integrity. Moreover, these EVs contained human gDNA sequences that corresponded to the xenograft cells, successfully identifying gDNA sequences associated with the biology of glioblastoma in all three types of EVs. Common sequences such as ERBB2, EGFR, CDK4, AKT3, and MDM2 were detected in all EVs [83,84]. However, IDH1 was specifically found in ABs and exosomes. Additionally, certain sequences were exclusive to specific types of EVs, for example, PIK3CA, MDM4, IDH2, and ASCL1 were unique to ABs, AKT1 was specific to microvesicles, and MGMT and RB1 were present only in exosomes. Consequently, the analysis of EV-derived DNA can serve as a biomarker for both high-grade and low-grade gliomas. This discovery is crucial in standardizing the use of EVs as a biomarker for brain gliomas, providing a less invasive method of detection compared to cerebrospinal fluid (CSF) analysis. This is particularly beneficial for patients with intracranial hypertension and could potentially lead to more effective and improved treatment of brain tumors.

Temozolomide (TMZ) is an oral alkylating agent commonly used for the treatment of glioblastoma. However, one of the primary challenges in using TMZ for glioblastoma treatment is inherent resistance [85,86]. Patients with MGMT fusion or low methylation of the MGMT promoter generally exhibit significantly higher MGMT expression, while those with high levels of DNA mutations tend to have the lowest MGMT expression [87,88]. Thus, MGMT promoter hypermethylation currently remains the only known biomarker for acquired TMZ resistance in glioblastoma patients. Barbara et al. [89] employed CRISPR/Cas9 technology to induce MGMT reordering in glioma cells, discovering that genomic rearrangement permits regulation by more active promoters, leading to increased MGMT expression and enhancement of TMZ resistance. Additionally, the fusion could be detected in exosomes derived from tumors, suggesting that exoDNA has the potential to serve as a biomarker for detecting MGMT fusions that confer TMZ resistance in vivo, thereby functioning as a recurrence biomarker.

Approximately one-third of patients with GBMs exhibit amplification of EGFR and its active mutant EGFRvIII, along with other mutations like MGMT and IDH, which play a crucial role in the development of the disease [88,90,91]. EGFRvIII activates multiple signaling pathways and possesses enhanced oncogenic and transforming properties, leading to increased tumor survival, invasiveness, stem-like characteristics, and angiogenesis [92]. Furthermore, it is associated with resistance to chemotherapy and radiation therapy, making it a specific biomarker for poor prognosis in GBM patients [93]. The presence of EGFRvIII has been linked to resistance to EGFR inhibitors, such as gefitinib and erlotinib [94]. Consequently, GBM patients expressing EGFRvIII often show limited response to these treatments, necessitating the exploration of alternative therapeutic approaches. Amplification of mutant DNA sequences in plasma indicates the presence of tumor-derived circulating DNA in the bloodstream, indicating incomplete tumor resection [90]. The detection of EGFRvIII DNA deletion mutations in EVs derived from peripheral blood holds promise as a method for using EVs EGFRvIII as a biomarker for anti-EGFRvIII vaccines and other targeted therapies. Additionally, the analysis of CSF-derived EVs demonstrated significantly higher levels of EGFRvIII-positive EVs compared to those expressing wild-type EGFR, further supporting its potential as a diagnostic biomarker [20,95,96,97]. To sum up, the use of exoDNA as a biomarker may assist in monitoring tumor progression, evaluating treatment responses, and guiding clinical decision making in glioma patients.

#### 3.2.3. HCC

Previous studies have confirmed that the c.747G>T mutation in TP53 is one of the most common mutations among HCC patients [98,99,100,101,102]. Yong et al. [102] successfully demonstrated the presence of the c.747G>T mutation in ctDNA and exoDNA of HCC patients, providing additional evidence for its significant role in the progression of HCC. HCC patients with high-frequency mutations (>67%) have a median recurrence-free survival (RFS) of only 63 days, which is significantly shorter than the 368-day median RFS for patients with low-frequency mutations (<67%). Importantly, even in patients with other favorable pathological features, such as lower alpha-fetoprotein levels (<400 ng/mL), the absence of cirrhosis, the absence of tumor thrombus, and solitary tumors, high-frequency mutations are still associated with a poorer prognosis [103,104]. Furthermore, there is an association between TP53 mutations and age with microvascular invasion (MVI), with tumors having high-frequency mutations more likely to invade microvessels. These findings suggest that the mutation frequency detected in exoDNA can serve as an independent risk factor for the prognosis of HCC, identifying patients with worse RFS in HCC and providing valuable information for the development of prognostic assessments and personalized treatment strategies.

#### 3.2.4. Neuroblastoma (NB)

Existing research supports the use of exoDNA as a non-invasive biomarker for the molecular diagnosis of NB [33]. The exoDNA of NB patients carries specific gene mutations unique to NB, including mutations in well-known oncogenes and tumor suppressor genes such as ALK, CHD5, SHANK2, PHOX2B, TERT, FGFR1, and BRAF [105], which aids in distinguishing NB and separating it from other forms of cancer. Furthermore, exoDNA from NB patients can be used to identify genetic mutations associated with acquired drug resistance, such as mutations in ALK, TP53, and RAS/MAPK genes observed in relapsed patients [106]. This information is crucial for devising personalized treatment strategies for these patients. Additionally, the number of single nucleotide variations (SNVs) detected in exoDNA is higher compared to SNVs obtained from tumor tissue biopsies of the same patient, a difference that suggests the SNVs in exoDNA originate from different tumor sites. This reduces the impact of spatial heterogeneity in NB and indicates that, even at lower frequencies, exoDNA can still capture parent tumor SNVs. Additional studies have shown that NB patients with a high tumor mutation load (TML) in exoDNA have a poorer prognosis compared to those with low TML values. This suggests that assessing the TML in exoDNA can provide valuable prognostic information for NB patients. Overall, exoDNA effectively captures the dynamic changes in tumors, and it can be isolated and enriched from plasma by targeting specific surface markers on EVs. This enables the rapid and convenient collection of exoDNA, making it a promising LB material. By utilizing exoDNA in LB, personalized drug treatments for children affected by NB can be advanced.

#### 3.2.5. Bladder Cancer (BC)

Currently, the process of diagnosing and monitoring BC often involves invasive procedures such as cystoscopy and cytological techniques, which carry risks of infection, pain, and hematuria [112]. Moreover, the existing FDA-approved biomarkers for cancer detection often exhibit inconsistent performance [113]. Therefore, it is crucial to continuously explore and identify more effective biomarkers that can reliably detect BC.

A study by Zhou et al. [107] compared the mutations associated with BC in exosomes found in the urine and serum of BC patients. The researchers observed that the concentration of exosomes in the urine of BC patients was significantly higher compared to samples from healthy people. There was also a trend of increasing exosome content in the serum of BC patients. Exosomes derived from healthy samples or serum from BC patients contained substantial amounts of exoDNA, which likely originated from tumor cells. This suggests that direct contact between tumor cells and urine, specific to BC, may play a role [108,109]. However, it is important to consider the limitations of the specific enrichment of biological fluids when drawing this conclusion. Further verification of the results obtained from comparing biological fluids (serum and urine) is needed to ensure reliability. Additionally, gDNA was detected in the exosomes of urine from BC patients but not in those from healthy samples. Whole exome sequencing of DNA extracted from urinary and serum exosomes, bladder tumor tissues, and normal tissues revealed that gene mutations common in BC (RXRA, TP53, FGFR3) that often occur in BC, as well as 30 UTR variants, in the DNA of exoDNA from the tumor tissue and urine [110,111]. Moreover, somatic driver mutations in BC-related genes were detected in the urine exoDNA analysis, demonstrating the uniqueness of exoDNA and its potential connection to the limitations of random biopsies and tumor heterogeneity.

In conclusion, the research results suggest that exoDNA can serve as an objective and non-invasive biomarker and target for BC. Combined sequencing of tumor biopsy DNA and urine exoDNA would provide a better representation of the genetic heterogeneity of tumors compared to individual small biopsy samples. These findings could potentially revolutionize the diagnosis and monitoring of BC, offering a less invasive and more accurate approach for detection and treatment.

## 4. Current Challenges and Future Perspectives of exoDNA

### 4.1. Challenges and Limitations

This review article focuses on the biogenesis, loading mechanism, and diagnostic potential of exoDNA in LB for tumor detection. However, there are still potential challenges in the current research and application of exoDNA (Figure 3).

Firstly, the cellular source of mutations in exoDNA remains unknown, making it challenging to identify the specific tissues or cells responsible for the mutations (Figure 3a). This limitation poses difficulties in performing early cancer screening and localizing organ invasion. To address this issue, it is suggested to use molecular protein markers to distinguish EVs originating from different tissue cells prior to DNA sequencing. By analyzing the DNA mutations in different EV subtypes, the genetic mutations in exoDNA can be correlated with their cellular origins. Future research should focus on identifying specific cell-surface protein markers of EVs to establish the correspondence between EVs and organ tissues.

Secondly, further research is needed to explore the clinical utility of exoDNA. Next-generation sequencing (NGS) of DNA samples requires a large volume of body fluids; however, there is the fundamental issue of a low concentration of exoDNA in plasma and a limited release of exoDNA in the early stages of tumors (Figure 3b). For instance, breast cancer screening currently requires testing approximately 150–300 mL of blood each time, which entails high costs and places a high demand on patients [114,115]. Implementing longitudinal continuous monitoring of exoDNA presents challenges in the clinical application of LB. Future efforts should focus on developing highly sensitive detection technology for exoDNA and applying advanced bioinformatics techniques to the use of LB with exoDNA to reduce the amount of plasma samples needed. Currently, technologies such as enhanced TAm-Seq (eTAm-Seq), deep methylation sequencing, CAPP-Seq, unique molecular identifiers (UMIs), target error correction sequencing (TEC-Seq), microfluidic chips, and others have demonstrated advancements in improving sensitivity and specificity [116]. Improving the detection sensitivity and specificity of plasma samples and enhancing the analytical performance of exoDNA are crucial for the future of exoDNA detection.

Thirdly, it is important to note that exoDNA mutations occur with a low frequency and are distributed differently in the three types of EVs (Figure 3c) [80]. This results in low efficiency for DNA sequencing. For instance, certain mutations associated with glioblastoma may exist in all three types of EVs, while other mutations may only be present in one or two subtypes. To enhance the rate of target DNA detection, differential centrifugation could be utilized to separate exosomes, ABs, and microvesicles. Subsequently, sequencing could be performed on subtypes containing target DNA mutations. This approach not only increases the detection rate of mutations but also reduces the cost of sequencing.

Fourthly, the tumor specificity of exoDNA is not uniform. Not all exoDNA is shed from primary or metastatic tumors, and it is unclear whether the detected changes accurately represent tumor heterogeneity (Figure 3d). Accurate identification of tumor-derived EVs can help reduce interference from non-tumor-derived exoDNA. Current studies have shown that tumor-derived EVs have an abundance of common membrane proteins such as GPC1 [71]. Future research should aim to determine the characteristic molecules of tumor-derived EVs, including other surface proteins, or differentiate them based on functional properties to improve the representativeness of DNA detection. Additionally, not all detected characteristic exoDNA mutations are related to cancer, as they can be confused with clonal hematopoiesis of indeterminate potential (CHIP) [117,118]. There is a higher incidence of CHIP in the elderly, and the incidence of tumors also increases with age. Therefore, it is necessary to differentiate tumor-related exoDNA mutations from gene mutations that occur in hematopoietic stem cells. Future approaches could focus on DNA sequencing and analysis of molecular markers to detect CHIP-specific gene mutations, such as DNMT3A, ASXL1, and ATM, to exclude non-tumor interference and improve the accuracy of LB.

Fifthly, tumor-derived exosomes are not uniformly shed from all sites and distributed (Figure 3e). Currently, most liquid biopsies are derived from blood samples; however, there should be further exploration of other body fluid sources, since existing studies indicate that fluids from different sites or types of tumors may have a higher biological enrichment of exoDNA. Examples include urine specimens from BC and NSCLC patients, pleural effusions from advanced lung cancer patients, ascites from epithelial ovarian cancer patients, CSF from lymphoma patients, and saliva from OSCC patients [118,119]. Therefore, multiple sample collections (plasma, urine, CSF) and repeated collections in combination with imaging examinations should be carried out to improve the exoDNA capture rate and detection sensitivity. Additionally, attention should be paid to standardizing the data across different sites and samples.

### 4.2. Therapeutic Applications

Due to its involvement in intercellular communication and its significant role in tumor development, exoDNA holds potential for therapeutic applications in gene therapy. For instance, leveraging the endogenous regulation of exoDNA to target and control gene expression in tumor cells can stimulate genes that promote apoptosis. The functional genes and regulatory elements carried by exoDNA can be utilized to overcome tumor cell resistance to drug therapy and hormone treatment, modulate immune cell function, and enhance cytotoxic effects. However, before designing therapeutic approaches, it is essential to gain a precise understanding of the mechanisms underlying exoDNA packaging. This includes comprehending the assembly patterns of exoDNA and the signaling pathways it affects.

## 5. Conclusions

In our comprehensive review, we systematically assessed the current research on exoDNA in LB and outlined potential challenges and future directions. The key areas of focus and challenges in exoDNA research are as follows: (1) identifying the cellular or tissue organ origins of exosomes; (2) understanding the distribution of exoDNA mutations among subtypes of EVs; (3) enhancing the sensitivity of exoDNA detection; (4) investigating the tumor specificity of exoDNA; (5) exploring alternative sources of LB beyond plasma; and (6) exploring the potential of tumor gene therapy based on exoDNA. By highlighting these factors, we aim to provide guidance for advancing the field of exoDNA research and avoiding potential blind spots and detours.

## Figures and Tables

**Figure 1 cancers-16-00057-f001:**
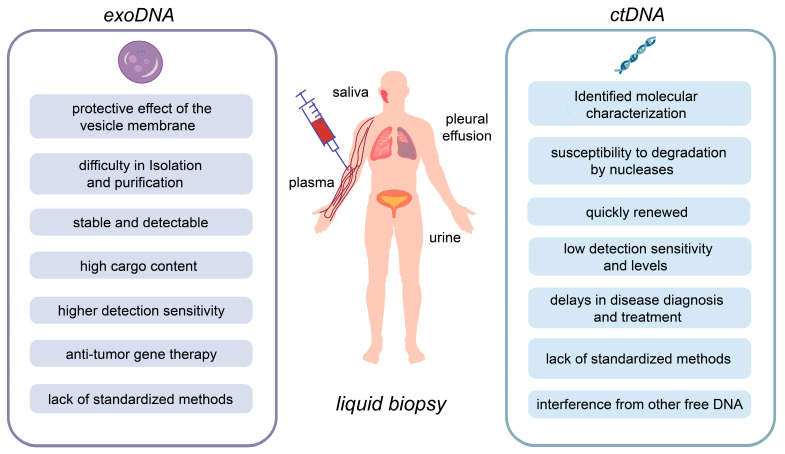
Comparison of exosomal DNA (exoDNA) and circulating tumor DNA (ctDNA) in liquid biopsy (LB). Schematic representing the advantages and disadvantages of exoDNA and ctDNA, both being components of circulating free DNA (cfDNA) in LB and typically detectable in serum or enriched through exosomes. Notably, exoDNA benefits from its protection by the exosome membrane, which enhances its stability and detectability, thereby compensating for the lower abundance of ctDNA in plasma samples.

**Figure 2 cancers-16-00057-f002:**
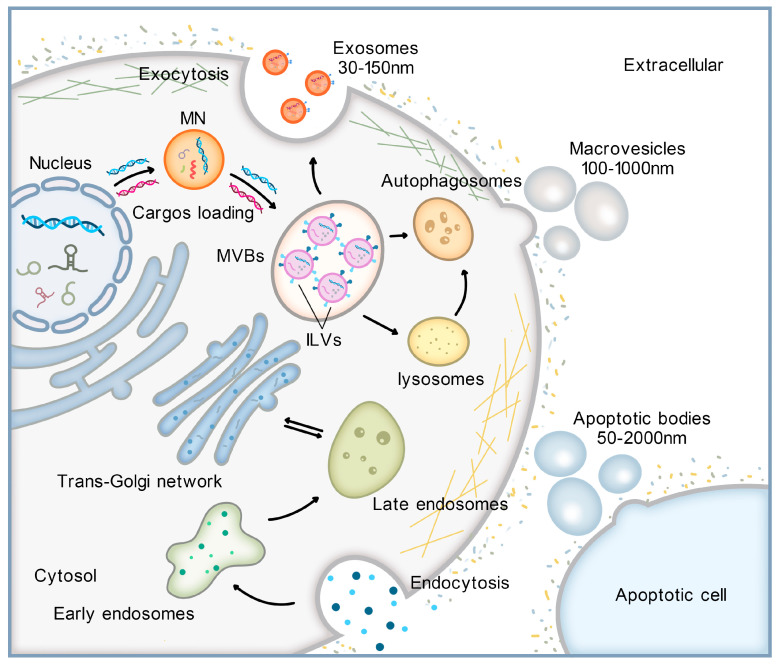
The biogenesis of extracellular vesicles (EVs) and exosomal DNA (exoDNA) loading mechanism. Exosomes originate from the process of endocytosis. The formation begins with the inward budding of the plasma membrane (PM), followed by the development of early endosomes, then late endosomes mature and undergo inward budding to form multivesicular bodies (MVBs). Eventually, MVBs fuse with the PM, allowing the intraluminal vesicles (ILVs) to be secreted as exosomes into the extracellular space. Microvesicles are formed when PM directly buds outwards. Apoptotic bodies (ABs) are generated during the late stages of programmed cell death and result from membrane blebbing. Micronucleus (MN) and MVBs are located near the cell nucleus. The membrane of MN is known to be fragile, and within it, cargo such as DNA is loaded into ILVs with the help of tetraspanins.

**Figure 3 cancers-16-00057-f003:**
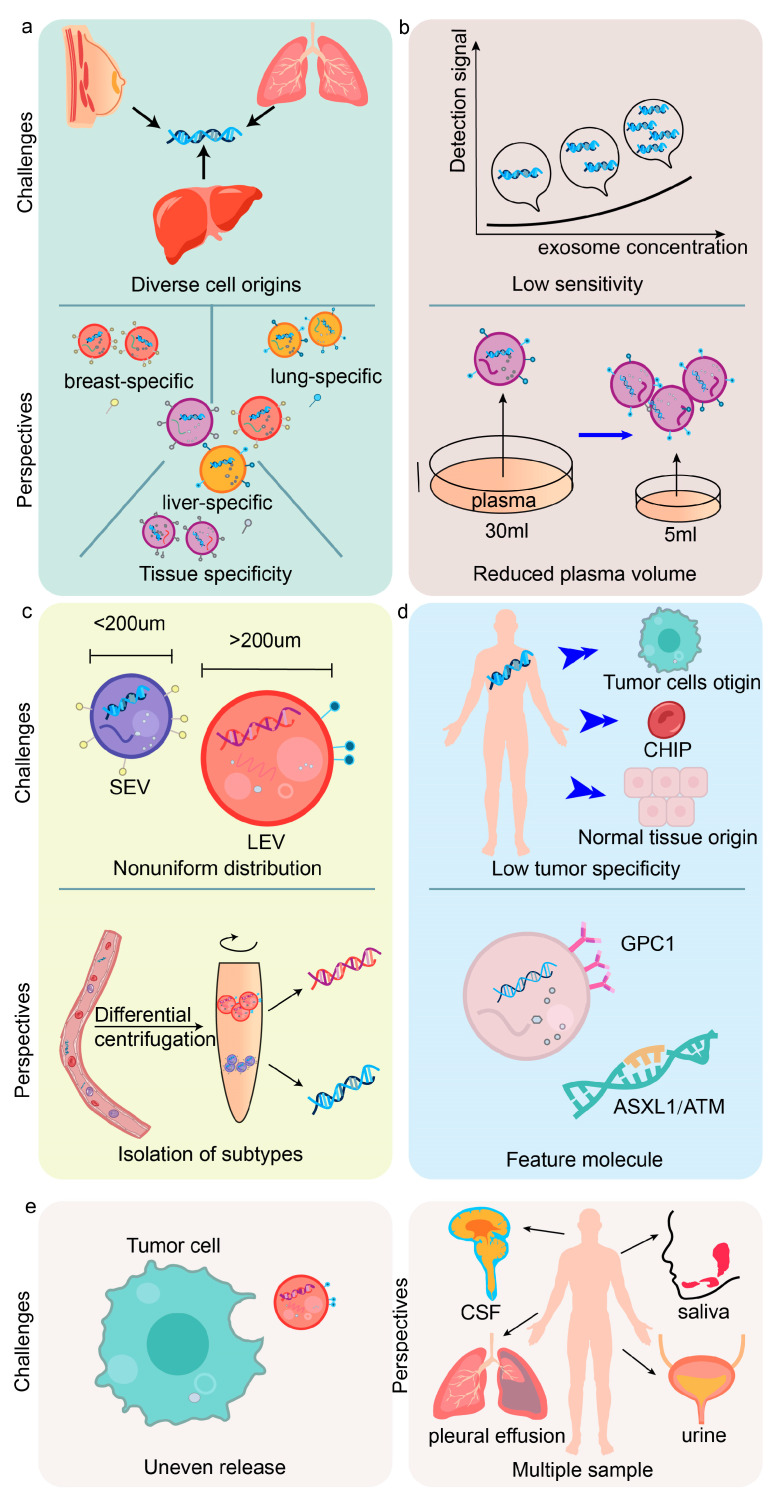
Challenges and future perspectives for the application of exosomal DNA (exoDNA). (**a**) It is not possible to distinguish the cell origin of exoDNA mutations, necessitating the differentiation of extracellular vesicles (EVs) from various tissue sources using molecular protein markers, among other methods. (**b**) exoDNA sequencing requires a substantial volume of bodily fluid, and to reduce the amount of plasma needed, the sensitivity of the detection technology should be enhanced. (**c**) exoDNA mutations exhibit different distribution patterns among the three types of EVs, therefore, differential centrifugation should be employed to separate and isolate them. (**d**) The tumor specificity of exoDNA is prone to interference, demanding the use of characteristic molecules to exclude clonal hematopoiesis of indeterminate potential (CHIP) and DNA originating from normal tissue sources. (**e**) The shedding and distribution of exosomes derived from tumors is not consistent across all sites. In the future, it will be important to investigate other fluid samples as potential sources of exosomes, and efforts should be made to enhance the rate of capture and detection sensitivity of exoDNA.

**Table 1 cancers-16-00057-t001:** List of known mtDNA mutations in exosomes and their clinical utility.

Tumors/Cell Line Types	Types of exoDNA	DNA Mutations/ Expression	Tissue Origins of exoDNA	Characteristics and Applications	Diagnostic/Treatment Roles	References
PCa	mtDNA	RCI mutations	plasma	circulating exosomes from PCa sera carried RCI-mtDNA and mt integrity associated proteins; co-localized with the mt in the PCa cells.	the pathogenic mtDNA mutation from RCI in circulating exosomes help early PCa detection, monitoring, and surveillance.	[24,36,37,38,39]
	Three-genes	depression	urine	EPI score > 15.6 can identify > GG2 when PSA level of 2–10 ng/mL	Assist screening for PCa, reducing unnecessary biopsy and treatment	[38,40,41,42,43]
ER^+^ breast cancer	mtDNA	copy number	plasma of patients, the conditioned media of cancer and stromal cell cultures		The transfer of ev-mtDNA serves as a carcinogenic signal leading to endocrine therapy resistance in OXPHOS-dependent breast cancer.	[44,45,46,47,48,49]
Glioblastoma	mtDNA	copy number changes and point mutations	tissue and plasma	mtDNA copy number in brain tissue and EV was lower in the glioblastoma patient group compared to the control group	mtDNA copy number changes as a biomarker for Glioblastoma	[50,51,52,53,54,55,56,57,58,59,60]
Ovarian cancer	mtDNA	copy number changes	whole blood/plasma	mtDNA copy number	mtDNA copy number changes can be used as a signal of ovarian cancer progression, and wb-mtDNA copy number can provide information about the appearance of early serous ovarian cancer	[61,62,63,64,65,66]

**Table 2 cancers-16-00057-t002:** List of known nuclear DNA mutations in exosomes and their clinical utility.

Tumors/Cell Lines Types	Types of exoDNA	DNA Mutations/ Expression	Tissue Origins of exoDNA	Characteristics and Applications	Diagnostic/Treatment Roles	References
PDAC	Nuclear DNA	KRAS, P53	plasma	Exosomes contain dsDNA, similar to the gDNA	Early stage cancer screening and detection purposes by identifying specific mutations	[69]
	Nuclear DNA	KRAS	plasma	GPC1^+^ crExos	PDAC’s biomarker	[71]
	Nuclear DNA	KRAS MAF	plasma	Longitudinal continuous biopsy monitoring of exoDNA	Predict the outcomes of adjuvant therapy for localized disease and the progression of metastatic disease	[6,70,72,73,74,75]
Glioma	exoDNA	concentration of exoDNA	plasma	concentration of plasma exoDNA is correlated with tumor volume and mitotic index	Detecting early recurrent high-grade gliomas and asymptomatic low-grade gliomas	[76,77,78,79]
	Nuclear DNA	IDH1G395A, ERBB2, EGFR, CDK4, AKT3, MDM2, PTEN, etc.	peripheral blood	All three types of EVs secreted by human glioblastoma cells can traverse the intact BBB. EvDNA can be used to detect IDH1G395A	The standardized use of EVs as biomarkers for brain gliomas, less invasive method of detection compared to CSF analysis	[80,81,82,83,84]
	Nuclear DNA	MGMT	plasma	exoDNA is a biomarker for detecting MGMT fusions	Detecting MGMT promoter hypermethylation for acquired TMZ resistance in glioblastoma patients	[85,86,87,88,89]
		EGFR, EGFRvIII	peripheral blood, CSF	Peripheral blood exoDNA mutations can reflect the status of tumors	EGFRvIII in exosome as a biomarker for anti-EGFRvIII targeted therapies	[20,88,90,91,92,93,94,95,96,97]
HCC	Nuclear DNA	TP53(c.747G>T)	serum	Independent risk factor for HCC prognosis	Identify the poorer RFS, provide prognosis personalized treatment	[98,99,100,101,102,103,104]
NB	Nuclear DNA	ALK, CHD5, SHANK2, etc.	plasma	exoDNA carries specific gene mutations that are characteristic of NB and associated with acquired drug resistance	Biomarkers for NB diagnosis and personalized drug treatments	[33,105,106]
BC	Nuclear DNA	RXRA, TP53, FGFR3, 30 UTR variants	urine and serum	RXRA, TP53, FGFR3, 30 UTR variants, in the DNA of tumor tissue and urine exoDNA	Combined sequencing of tumor biopsy DNA and urine exoDNA can provide a better representation of the genetic heterogeneity of tumors	[107,108,109,110,111]

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
