# Peer review of "Exosomal DNA: Role in Reflecting Tumor Genetic Heterogeneity, Diagnosis, and Disease Monitoring"

_cancers, 2023, doi:10.3390/cancers16010057_

Round 1
Reviewer 1 Report
Comments and Suggestions for Authors
My major comments are.
1. What is LB?. Please mention fill name and put short form in bracket.
2. Line 24-26, By shedding light on 24 both known and unknown aspects of exoDNA, this review aims to guide the field of LB, avoiding 25 unnecessary blind spots and detours. This sentence, looks a bit comples to understand, please simplify it.
3. On line 48, you used words like 'DNA species.' provide a reference for this or correct it.
4. Please try rewriting lines 52 to 54 as their last line is not matching the paragraph.
5. In the introduction section, additional information related to the EVs needs to be presented. Explaining the types of EVs and the differences between them will make it easier to understand EVs for those who are not familiar with them. Please cite these reference. https://www.mdpi.com/2227-9059/9/10/1373
https://www.mdpi.com/1422-0067/23/17/10010
https://link.springer.com/article/10.1186/s12882-021-02417-8
https://ojrd.biomedcentral.com/articles/10.1186/s13023-020-01607-1
6. In the introduction section, please avoid using words repeatedly; Avoid duplicating the same lines as the abstract part.
7. The section on 'Formation of EVs and mechanisms of exoDNA loading' is written superficially without mentioning any proteins involved in formation or cargo loading. To easily understand their mechanism of action, it is necessary to mention the proteins involved.
8. In the section concerning Research Progress on the Application of exoDNA in LB, references are missing in lines 124, 138, 141, 143, 147.
9. No reference has been given for the part about “MtDNA as a potential biomarker for tumor diagnosis and treatment”, and there is quite a bit of confusion in the paragraph.
10. It is necessary to write tables 1 and 2 with borders around them to avoid confusion.
11. "Prostate cancer (PCa)" paragraph is written without references, please includes references for these.
12. In lines 307 to 343, sudden references have been inserted that are causing a lot of confusion in this paper
13. "There are many shortcomings throughout the paper, such as repetition of several words, repeating the same line multiple times, and references are missing in almost every paragraph as pointed out in previous comments. The quality of the images is very poor.
14. The figure 2 is showing just biogenesis of exosomes, I would suggest to show the biogenesis of microvesicles also in this figure. Because, both exosomes and microvesicles are types of EVs.
15. In table 2, references should be inserted following journals guidelines. References in table 2 are meshing up everything.
16. Line 307 to 343. It doesn’t make any sense to includes references like this.
17. The conclusion part needs to be summarize well.
Comments on the Quality of English Language
Extensive English editing is required.
Author Response
We would like to thank the three reviewers for their positive comments on the scientific content of our work. The comments and suggestions made by the reviewers are very helpful for us to revise the manuscript. Detailed reply to the comments and suggestions has been made as follows. (Note: The reviewers’ comments are highlighted here in blue, and our responses are highlighted in red).

Reviewer 2 Report
Comments and Suggestions for Authors
Thi review is mostly well-written, with clear descriptions and references.
i suggest to the authors to introduce a brief paragraph regarding the knowledge of the content of exosome including which type of proteins and RNA (for example miRNA) because this, together with exoDNA, has a strong impact on the diagnosis, heterogeneity and disease monitoring.
Minor issue: some abbreviations are cited before their explanation: For example: EVs line 33 is explained in line 41. On the contrary in table 2, Pancreatic cancer can be substitued with PDAC.
In Table 2, we find: Error! Bookmark not defined.
Comments on the Quality of English Language
the quality of English is good
Author Response

(The authors gave the same response as above.)

Reviewer 3 Report
Comments and Suggestions for Authors Review "Exosomal DNA: Role in Reflecting Tumor Genetic Heterogeneity, Diagnosis, and Disease Monitoring" is an interesting study on the function of exosomal DNA in various most common cancers. This review has not only a scientific aspect, but also a clinical one. The developed issues can be used in the diagnosis and monitoring of the disease. It also has a potential predictive aspect. The authors included several figures, which is very important when reading this review. I propose to re-place explanations of abbreviations under the figures, even though they were explained in the abstract and introduction. References contain articles published in the last 10 years. This proves the topicality and importance of the issues discussed. Due to the significant scientific and clinical importance of the discussed prognostic role of exosomal DNA in cancer, I propose that the review "Exosomal DNA: Role in Reflecting Tumor Genetic Heterogeneity, Diagnosis, and Disease Monitoring" be published in CANCERS in its current form.Wyniki tłumaczenia
Tłumac
Comments on the Quality of English Language Minor editing of English language is required.Wyniki tłumaczenia
TłumacMinor editing of English language required
Author Response

(The authors gave the same response as above.)
